# Learning to Reason via
# Program Generation, Emulation, and Search

**Nathaniel Weir**[*][†]
Johns Hopkins University
nweir@jhu.edu

**Muhammad Khalifa**[*][†]
University of Michigan
khalifam@umich.edu

**Linlu Qiu**[*]
MIT
linluqiu@mit.edu

**Orion Weller**[*]
Johns Hopkins University
oweller@cs.jhu.edu

**Peter Clark**
Allen Institute for AI
peterc@allenai.org

## Abstract

Program synthesis with language models (LMs) has unlocked a large set of reasoning abilities; code-tuned LMs have proven adept at generating programs that solve a wide variety of algorithmic symbolic manipulation tasks (e.g. word concatenation). However, not all reasoning tasks are easily expressible as code, e.g. tasks involving commonsense reasoning, moral decision-making, and sarcasm understanding. Our goal is to extend an LM's program synthesis skills to such tasks and evaluate the results via *pseudo-programs*, namely Python programs where some leaf function calls are left undefined. To that end, we propose, **Co**de **G**eneration and Emulated **EX**ecution (CoGEX). CoGEX works by (1) training LMs to generate pseudo-programs, (2) teaching them to *emulate* their generated program's execution, including those leaf functions, allowing the LM's knowledge to fill in the execution gaps; and (3) using them to search over many programs to find an optimal one. To adapt the CoGEX model to a new task, we introduce a method for performing program search to find a single program whose pseudo-execution yields optimal performance when applied to all the instances of a given dataset. We show that our approach yields large improvements compared to standard in-context learning approaches on a battery of tasks, both algorithmic and soft reasoning. This result thus demonstrates that code synthesis can be applied to a much broader class of problems than previously considered.[1]

## 1 Introduction

Recently there have been rapid advances in training language models (LMs) to generate code rather than natural language (NL), following the intuition that code may be more effective than NL for certain tasks, such as those requiring complex calculations, iteration, or data structure manipulation(Chen et al., 2022; Gao et al., 2023). Although successful, these works have mostly studied tasks conducive to a programmatic paradigm, such as symbolic manipulation or algorithmic reasoning, i.e., tasks for which a clear compilable program can be devised. However, it is unclear how to apply this approach to "softer" reasoning tasks such as commonsense and social reasoning, where algorithmic solutions are less obvious (Zhang et al., 2023a).

---

[*]Work done in part during internships at Allen Institute for AI.

[1]Our released dataset, fine-tuned models, and implementation can be found at https://github.com/nweir127/CoGEX.

[†]Co-first authors.

38th Conference on Neural Information Processing Systems (NeurIPS 2024).

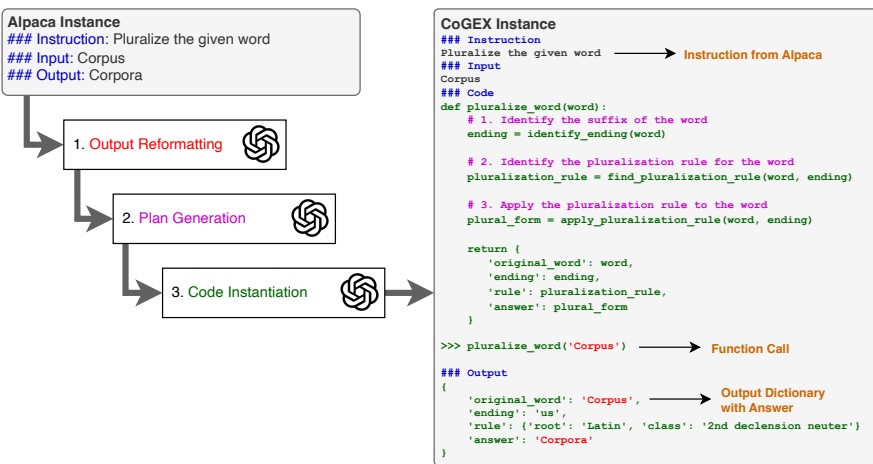

Figure 1: Example from the CoGEX dataset automatically converted from an Alpaca (Taori et al., 2023) instance via LLM prompting. We train the model to receive the instruction and input and generate the Python program and function call (as an intermediate), before outputting the final dictionary that contains the answer and any intermediate reasoning steps.

Our goal is to expand an LM's program synthesis skills to such softer reasoning tasks. Our approach builds on the insight that, beyond generating code, LMs can also *emulate the execution* of code. This includes handling function calls that are defined only by name and documentation, even if they lack a full implementation. We refer to these types of programs—where only the function skeletons are provided without actual code implementations—as *pseudo-programs*. Such pseudo-programs can encompass both well-defined reasoning steps, such as mathematical or algorithmic operations, as well as function calls representing less precise reasoning, such as commonsense logic. This work investigates whether generating and pseudo-executing such programs can effectively address soft reasoning tasks in addition to traditional algorithmic problems.

To achieve this, we propose a novel approach: training models to follow NL instructions by generating a program and then *emulating* that program's code execution. Our paradigm, called CoGEX, changes the inference process to (**1**) generate a Python function given an arbitrary instruction and optional input, (**2**) generate a call to that function, and (**3**) produce the result of simulating its execution. Unlike other work (Zhang et al., 2023b; Li et al., 2023), we do not use a Python interpreter to execute any code; rather, the model is trained to emulate execution. This allows the generated code to deliberately include calls to underspecified functions (i.e., where only the function name and documentation are included), by allowing the LM emulator to *fill in* the functions implementations using its latent knowledge. We train the model to not only output the result of pseudo-executing the code but also the results of intermediate function calls in the code. CoGEX thus suggests a way to leverage the flexible ad-hoc reasoning abilities of LMs while encouraging programmatic reasoning via the program structure. We train CoGEX models by adapting the recent Alpaca instruction tuning dataset (Taori et al., 2023) into a set of analogous Pythonic examples by prompting GPT-4 to perform the conversion, and then use the resulting CoGEX dataset to fine-tune smaller (7B and 13B) LMs to answer instructions via code.

The CoGEX paradigm allows us to explore a new way to learn new tasks: identifying a *general* program that applies across a task, such that new task instances can be solved by emulating calls to that one program. Inspired by work on hard prompt tuning (Wen et al., 2024) and example selection for in-context learning (Gupta et al., 2023a), we introduce a search procedure that uses the frozen CoGEX model to try out many programs on a set of training examples and identify which single program is optimal for the dataset. The procedure, termed CoTACS: **Co**GEX **T**ask **A**daptation via **C**ode **S**earch, performs no parameter updates and requires saving nothing more than a program string.

We evaluate over a diverse suite of reasoning tasks, including commonsense QA, text classification, and math datasets. These datasets cover symbolic tasks that could conceptually benefit from programmatic operations and standard natural language tasks whose solutions might not be easily described in code. We find that applying CoTACS leads the CoGEX models to substantially outperform the comparable NL-based LM using the same original checkpoint and the same set of training examples

available for in-context learning, even in the few-shot regime. CoTACS thus gives us one way to fit a model to a new dataset without having to perform any gradient descent or parameter updates, both for algorithmic and softer reasoning datasets. Our contributions are thus:

1. A novel reasoning paradigm, CoGEX, that trains language models to generate and emulate the execution of pseudo-programs. CoGEX is a general paradigm that allows LMs to leverage code for different types of reasoning.

2. A program search method, CoTACS, enabling a task-general program suitable for a dataset (rather than a single instance) to be found using a CoGEX model.

3. A dataset, derived from the Alpaca instruction tuning dataset, for training CoGEX models.

Overall, this work provides a significant step in showing how code generation can be applied to a much broader class of problems than previously considered.

## 2 Approach

In this section, we start by formalizing our approach and describing our data construction process (§2.1). We then describe our program search approach to tune a CoGEX model on a given task through program search (§2.2).

### 2.1 Method: CoGEX

**Formulation.** Our goal is for the model to execute a given task by simulating code execution. That means our model will take as input the task description, generate a Python program, and simulate the expected output of executing that program. Formally, given a natural language (NL) task description $I$, optional input argument $A$, Python function $F$, function call $C$, and output dictionary $O$ designating the output from the program pseudo-execution, the LM will take $\langle I, A \rangle$ as input and generates $\langle F, C, O \rangle$ as output. Since the process is sequential, CoGEX models can work as either a reasoner $f_{\text{reasoner}}(I, A) \rightarrow (P, C) \rightarrow O$ or as a call-instantiating and execution-emulating model $f_{\text{emulator}}(I, A, P) \rightarrow C \rightarrow O$ that takes a pre-specified program $P$ and applies it to the variable arguments $A$. This latter formulation enables searching over the space of task-specific programs: searching for one $P_{\text{task}}$ to solve a class of problem (e.g., emotion classification) and then applying $f_{\text{emulator}}(I, A_i, P_{\text{task}})$ to emulate its execution on each instance $A_i$ of that problem. We expand on program search in §2.2.

**Training Data Construction.** As we want a general-purpose dataset that spans tasks with diverse reasoning requirements, we choose the Alpaca instruction tuning dataset (Taori et al., 2023). Following Peng et al. (2023), we rely on GPT-4[2] to convert the Alpaca dataset into their CoGEX versions. Specifically, every NL instance in the Alpaca dataset is mapped into a corresponding CoGEX version. We split the conversion process into three steps, each of which involves prompting GPT-4 with the output from the previous. This stepwise approach proved more effective than directly prompting GPT to convert each instance to code in one shot.

As depicted in Figure 1, the three steps are: **(1)** converting the outputs and (optional) inputs into Pythonic data structures like strings, lists, and integers whenever relevant as determined by GPT-4; **(2)** generating an instruction-specific plan, or a series of NL steps that should perform the task for any potential input; **(3)** instantiating the plan as a Python program whose inline comments are the plan steps and whose output is a dictionary containing all intermediate and final outputs that the LM believes would result from executing each step. Prompts for all steps can be found in Appendix A.

Importantly, we allow for GPT to include *undefined* functions, e.g., `identify_ending()` and `find_pluralization_rule()` in Figure 1. The goal is to leverage the LM knowledge to fill in the semantics of these undefined functions when emulating the execution of a given program. In addition, we include the program's *intermediate results* in the output dictionary before the final answer to encourage the model to stick to the NL reasoning plan delineated in the program comments. After defining the program, we cue GPT-4 to *call* the function on an argument, e.g. `pluralize_word('corpus')` which can reflect the optional Alpaca example input, or can reflect specific details from the instruction itself. Our prompts encourage GPT-4 to write a program that

---

[2]The dataset was constructed between August 7th–26th, 2023 using the `gpt-4` model in the OpenAI API.

is as general purpose as possible and not tied to a specific input: e.g. `pluralize_word(word)` is preferable to `pluralize_corpus()`.

Fine-tuning any LM on the resulting CoGEX dataset creates our desired model, which accepts any task description/input combination and responds by dynamically generating a Python program and then emulating its execution.

## 2.2 Program Search: CoTACS

A CoGEX model can generate a new program for any new task instruction and instance; however, some programs might be more or less effective at performing the task. How can we find the optimal program for a specific task, especially when some training data is available? As CoGEX relies on argument-accepting pseudo-programs, it naturally enables program optimization. Given multiple examples of a new task, we can search for *one* program that performs well, and then apply the same program to new test examples by invoking the program with different input arguments.

Our search process, CoTACS: **Co**GEX **T**ask **A**daptation via **C**ode **S**earch, finds a single program that optimizes a CoGEX model for a particular task dataset, enabling adapting a CoGEX model to a given task without learning any weight parameters. We learn a new dataset simply by using a finetuned CoGEX model to generate and then evaluate many program candidates to find the one that best fits the given dataset. As described in algorithm 1, we split a dataset $D$ of argument and output pairs $(a_i, o_i)$ into a small training set ($n = 300$ in experiments) and a larger development set; we then generate a separate code candidate for every training item and retain the programs with decent performance on the training set.[3] We then rank these programs according by their performance on the development set. For certain tasks, we find it beneficial to find *multiple* programs for a task and then at test time take a majority vote across the CoGEX model's responses using each code. To accomplish this, we retain some top-$k$ performing codes over the development set.

---

**Algorithm 1:** CoTACS search that identifies a set of $k$ programs $P_D$ that best adapts a CoGEX model to new dataset $D$

---

**Input:** CoGEX model $f$, Dataset $D = \{(a_1, o_1), (a_2, o_2), \ldots\}$, Instruction $I$, number of code candidates $n$, minimum training performance $\alpha$, task metric $\delta$
**Result:** Optimal programs $P_D$ that maximize model performance on $D$

---

1   Programs $\leftarrow \emptyset$;
2   TrainSet $\leftarrow$ RandomSample$(D, n)$;              `// Sample from` $D$
3   DevSet $\leftarrow D \setminus$ TrainSet ;         `// Remaining` $|D| - n$ `examples serve as dev set`

4   **for** $(a_i, o_i)$ in TrainSet **do**
5      |   $p_i \leftarrow f(I, a_i)$;              `// Sample a program for the instance`
6      |   TrainPerf $\leftarrow$ Evaluate$(p_i,$ TrainSet$)$;
7      |   **while** TrainPerf $< \alpha$ **do**
8      |     |   $p_i \leftarrow f(I, a_i)$ ;            `// Resample code if low performance`
9      |     |   TrainPerf $\leftarrow$ Evaluate$(p_i,$ TrainSet$)$;
10     |   **end**
11     |   Add $p_i$ to Programs;
12   **end**

13   $P_D \leftarrow \operatorname{argmax}_{P = \{p_1, \ldots p_k\} \subseteq \text{Programs}} \sum_{i=1}^{k}$ Evaluate$(p_i,$ DevSet$)$;
14   **return** $P_D$;

15   **Function** Evaluate$(p, D)$:
16     |   **for** $(a_i, o_i)$ in $D$ **do**
17     |     |   $(c_i, \acute{o}_i) \leftarrow f(I, a_i, p)$;         `// Run the model with program` $p$
18     |   **end**
19     |   **return** $\frac{1}{|D|} \sum_{i=1}^{|D|} \delta(\hat{o}_i, o_i)$;       `// Average task metric (e.g., exact match)`

---

[3]To ensure quality of the retained programs, a user-defined threshold $\alpha$ is used to keep only the programs whose training performance is at least $\alpha$. If no program achieves $\alpha$ after a fixed number of trials, we use the best performing sampled so far.

Table 1: Benchmark results by CoGEX models optimized for each dataset using the CoTACS method, compared to the corresponding off-the-shelf Llama-2 checkpoint performing 2-shot reasoning using a BM25 retrieval index of 1000 exemplars. Results are also compared to a zero-shot Alpaca model fine-tuned from the same checkpoint. The top score per size is **bolded**. Colored cells indicate changes ( gains , losses , or the  same ) relative to the best-performing non-CoGEX baseline (Alpaca or 2-shot). Results show that CoGEX with CoTACS outperforms the baselines on nearly every task and often does so even with only 10 examples.

| | | | Classification | | | Symbolic | | Math | | Commonsense | | |
|---|---|---|---|---|---|---|---|---|---|---|---|---|
| | | $N_{\text{train}}$ | CoLA | Emotn | SST | Coin | WSort | Sum | SVAMP | CSQA | SIQA | Avg |
| Alpaca 7B | 0-shot | 0 | 70.6 | 53.4 | 87.3 | 49.5 | 40.0 | 21.6 | 25.7 | 46.8 | 54.1 | 49.9 |
| Llama-2 7B | 2-S BM25 | 1000 | 57.5 | 55.2 | 82.1 | 32.4 | **45.5** | 35.2 | 34.7 | 45.7 | 46.0 | 48.3 |
| **CoGEX** Llama-2 7B | CoTACS $k=1$ | 10 | 75.0 | 52.2 | 86.9 | 50.8 | 40.6 | 61.3 | 33.3 | 42.3 | 50.1 | 56.7 |
| | CoTACS $k=1$ | 1000 | 78.5 | **56.2** | **91.2** | 60.0 | 39.5 | 62.8 | 41.3 | **52.7** | 57.7 | 60.0 |
| | CoTACS $k=3$ | 1000 | **79.2** | 56.0 | 90.9 | **61.9** | 40.8 | **63.6** | **42.7** | 52.4 | **59.3** | **60.8** |
| Alpaca 13B | 0-shot | 0 | 74.6 | 53.0 | 86.0 | 63.8 | 50.3 | 44.5 | 37.3 | 62.2 | 63.4 | 59.5 |
| Llama-2 13B | 2-S BM25 | 1000 | 78.9 | 54.0 | **92.4** | 48.6 | 50.0 | 38.7 | 46.0 | 63.4 | 62.8 | 59.4 |
| **CoGEX** Llama-2 13B | CoTACS $k=1$ | 10 | 80.9 | 55.1 | 88.9 | 58.0 | 51.3 | 61.3 | 42.8 | 59.2 | 57.6 | 61.7 |
| | CoTACS $k=1$ | 1000 | 81.0 | 56.4 | 92.1 | 65.7 | 50.8 | **64.0** | 48.3 | 63.4 | 64.6 | 65.1 |
| | CoTACS $k=3$ | 1000 | **81.0** | **56.6** | 92.1 | **69.5** | **51.6** | 63.9 | **50.3** | **64.0** | **65.5** | **66.1** |

## 3 Experiments and Results

Below we describe the training and evaluation of CoGEX models. We first show overall performance across a wide array of tasks compared to off-the-shelf baselines (§3.2) then ask a series of follow-up research questions investigating ablated scenarios (§3.3).

### 3.1 Experimental Setup

**Model Training.** We fine-tune the 7B and 13B variants of Llama-2 (Touvron et al., 2023). We use parameter-efficient training via Low-rank adaptation (LoRA) (Hu et al., 2021) with a rank of $r = 16$, dropout rate of 0.05, and LoRA weights added to the query, key, value, and output matrices in all layers. We train all models for five epochs using a batch size of 32 and a learning rate of 0.0003. As a validation set, we randomly sample 2K examples from the training set and keep the checkpoint with the lowest perplexity on the validation set for testing. Model training was done on a server with 128GB of RAM and 2 Nvidia A6000 48GB GPUs. On our dataset, training a single 7B model and 13B models took around 12 and 20 hours, respectively. To ensure a fair comparison, all the baselines are trained with the exact same hyperparameters.

**Datasets.** We measure CoGEX model performance on a variety of benchmarks ranging from symbolic manipulation to commonsense and social reasoning. We choose these datasets as representatives of tasks that involve complex reasoning and tasks whose solutions cannot be easily described in code. As for symbolic and math reasoning, we use the Word Sorting task from BIG-bench hard (Srivastava et al., 2022; Suzgun et al., 2022), the math word problem dataset SVAMP (Patel et al., 2021), the coin flip tracking dataset from Wei et al. (2022), and the large number arithmetic task (referred to as Sum) from Zelikman et al. (2022). For the last, we use the 5-digit examples for training and 6-digit for testing. We measure string-normalized exact match accuracy for all tasks. Following Zhang et al. (2023b), we evaluate on a series of **Text Classification** datasets: CoLA (2 labels) (Warstadt et al., 2019), Emotion (6 labels) (Saravia et al., 2018), and SST2 (2 labels) (Socher et al., 2013). We also evaluate on the **Commonsense Reasoning** datasets CommonsenseQA (Talmor et al., 2019) and Social IQa (Sap et al., 2019), which are 4- and 5-way multiple-choice datasets. We hand-write the instruction $I_{\text{task}}$ for these datasets as they do not provide any.

**Code Search.** For all datasets, we use a maximum of 1000 training examples. We use $n = 300$ training items to generate candidate codes and evaluate them on the remaining 700 items to identify the most generalizable ones. We experiment with retaining the top-$k \in \{1, 3\}$ performing programs for use at test time. For $k = 3$, we take a majority vote of answers, breaking ties randomly. We use

Table 2: Difference in performance by CoTACS $k = 3$ comparing Llama-2 vs Code Llama CoGEX models ('+' implies Llama-2 better). We see that Code Llama is more effective for some tasks but worse on others, while the 13B version performs worse than the 13B Llama-2 on all but 2 tasks.

| Model Size | CoLA | Emotn | SST | Coin | WSort | Sum | SVAMP | CSQA | SIQA |
|---|---|---|---|---|---|---|---|---|---|
| 7B | +1.3 | -0.8 | +1.2 | -10.5 | -4.5 | +6.2 | +2.7 | -3.6 | +4.2 |
| 13B | +0.5 | +0.2 | +0.9 | -12.4 | +2.4 | -9.3 | +3.0 | +6.8 | +4.7 |

sampling temperature $t = 0.7$ when generating candidate codes and $t = 0.05$ to generate answers given a program and argument. We report results on the released dev sets of all considered tasks.

**Baselines.** We consider two baselines that represent standard practices for adapting an LM to a new task: (1) few-shot prompting using the off-the-shelf Llama-2 and CodeLlama models and (2) zero-shot prompting using the Llama-2 models instruction-tuned from the original Alpaca dataset.[4] For the in-context learning baseline for (1) we use the same 1000 training data-points as CoTACS and optimize the examples by retrieving the most similar few-shot examples using BM25. For zero-shot alpaca models, we use the standard Alpaca-7B and -13B models. As CoTACS might not require many training examples to achieve strong performance, we compare the 1000-example CoTACS run with one that only uses 10 total examples to generate and evaluate candidates.

## 3.2 Results

Our main results depicting the difference in performance between CoGEX models tuned via Co-TACS versus off-the-shelf few-shot baselines and Alpaca models are shown in Table 1. The CoTACS method with 1000 training examples outperforms the baselines for a large majority of tasks and models (8/9 tasks for both Llama-2 7B and 13B). CoGEX shows particularly strong gains over the baselines in the Sum and coin flip tracking tasks (+10-20%) as expected due to its code-related nature. We observe that the CoTACS method with $N_{train} = 1000$ training examples performs best on average across the 9 tasks, and still performs better than the baselines with only $N_{train} = 10$ examples. Retaining the top $k = 3$ programs instead of 1 improves performance in most cases (+1% average).

**Instruction Following.** As our models are trained on instruction following in code, can they still perform instruction-related tasks as well as models trained on text-only Alpaca? We verify this by using `alpaca-eval` to compare Alpaca-7B against our CoGEX-7B model trained from the same base Llama model. We find a similar win rate (50% within the 2 SD range) indicating similar instruction following ability. Thus we can see that training on code-based instructions does not hurt standard instruction-following abilities, while opening up many possibilities for program search.

**Effect of Code Pre-training.** As we are fine-tuning LMs on code data and then evaluating them on tasks that are more or less code-related, a natural question to ask is whether LMs pre-trained on code datasets yield stronger CoGEX models. We investigate this by fine-tuning Code Llama (Roziere et al., 2023) on the CoGEX dataset instead of Llama. Table 2 shows the resulting change in performance using CoTACS ($k=3$) program search. The Llama-2 models show improved performance on Social IQa (+4%) but much worse on coin flip tracking (-10-12%). These results do not provide conclusive evidence that Llama-2 models are better or worse than Code Llama on particular task categories.

## 3.3 Ablation Studies

Here we present a series of ablation studies to ask the following questions:

**How many training examples are needed for search?** In the above experiments, we chose 1000 training examples and 300 program candidates for the CoTACS algorithm. This raises the question: how many examples are required to yield the strong performance provided by the search? We investigate this by simulating the algorithm and sampling 1000 trials with varying numbers of training examples and program candidates. Results are shown in Figure 2. In nearly all cases, performance with 50 or 200 training examples is within a couple of points of the full performance with the 300/1000 configuration. The performance when sampling 10 code candidates (green) lands within 2 points of sampling 300 candidates on 5 of the 7 datasets. Benefits do not appear on Word Sorting, as

---

[4]We also experimented with few-shot prompting the Alpaca models, but found them to perform significantly worse than zero-shot, which is likely a side effect of instruction tuning.

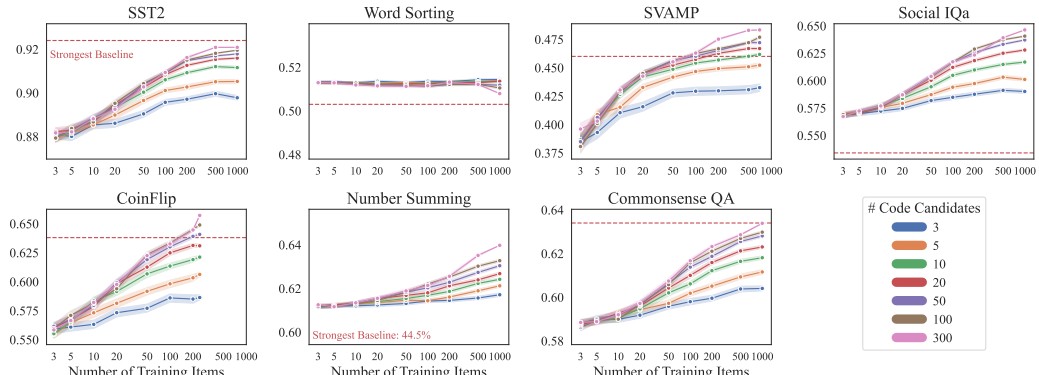

Figure 2: Change in CoTACS performance using Llama-2 13B as we increase the number of training items from 5 to 1000 and program candidates from 3 to 300. Results are averaged across 1000 trials.

performance lands between 0.51 and 0.515 regardless of configuration. This suggests that the range of quality in generated programs for the Word Sorting dataset is much smaller than the others, so picking between just a few candidates is sufficient. Overall, we see that we can significantly reduce the search space and still see large gains on most tasks.

**Is it better to execute an NL plan instead of Python code?** We have proposed a mechanism to generate Python programs whose steps are meant to reflect the reasoning process, stated in NL, that answers a given instruction. Is the code necessary, or can a model be trained to generate only the plan and achieve the same performance? We fine-tune a Llama model on a version of the CoGEX 52k-item training set where each Python program has been replaced with just the NL steps (removing step 3 of Figure 1). This **NL Plan** model still returns the same output dictionary with intermediate results. To fit the plan-only model to a dataset, we run the CoTACS algorithm but sample and retain the NL plans instead of programs. We see in Figure 4 (orange vs gold) that NL Plan CoTACS can match the performance of regular program CoTACS on some tasks (CoLA, SST, Word Sorting, Emotion), but performs much worse on others, particularly Coin Flip, SVAMP, and Sum. This follows the intuition that these tasks benefit from a programmatic reasoning paradigm.

**Is it better to find one program or generate a new one for each instance?** CoTACS finds one or multiple programs that can be reapplied to all task instances in a dataset to achieve high performance. Is this better than letting the CoGEX model generate a separate program for every instance? It might be the case that the latter allows for catering the program to the specifics of a particular task instance– e.g. in Figure 3, where the left (red) CoGEX-generated program has steps specifically crafted to identify actions to be taken by a particular person. Finding a single program disallows this flexibility. We investigate this question by running the model end-to-end on each instance. The CoTACS model performs the mapping $f(I_{\text{task}}, A_i, P_{\text{CoTACS}}) \rightarrow C_i \rightarrow O_i$ for each task instance $A_i$, while the end-to-end model performs $f(I_{\text{task}}, A_i) \rightarrow (P_i, C_i) \rightarrow O_i$. We sample from the latter using temperature $t = 0.05$.[5] Results are shown in Figure 4 (maroon vs gold); end-to-end performance is comparable to CoTACS only on Word Sorting and Sum. In all other cases, it is substantially worse.

---

[5] Increasing $t$ and/or using self-consistency (Wang et al., 2023) did not meaningfully affect performance.

```
def answer_question(question, options):
    """
    Answer a Social Interaction QA question.
    Args:
        question (str): the Social Interaction QA question.
        options (list): a list of potential answers.
    Returns:
        A `dict` containing the field 'answer', whose value is of type `str` and contains
        the letter of the correct answer, plus fields for intermediate steps of the reasoning process.
    """
```

**Item-Specific Program:**
```
# Step 1: Identify the key elements in the question.
key_elements = identify_key_elements(question)

# Step 2: Analyze the context of the question to
understand what action Sasha needs to take.
action = analyze_context(question, key_elements)

# Step 3: Return the letter of the correct option.
answer = select_option(options, action)

return {'key_elements': key_elements, 'action':
action, 'answer': answer}
```

**General Program:**
```
# Step 1: Identify the key elements in the problem.
key_elements = identify_key_elements(question)

# Step 2: Analyze the context to understand the relationship
between the key elements.
context = analyze_context(question, key_elements)

# Step 3: Based on the context, determine the likely answer.
answer = determine_likely_answer(context, key_elements, options)

return {'key_elements': key_elements, 'context': context,
'answer': answer}
```

```
>>> answer_question("Casey gave some money to Jesse so she could go to the movie. How would you describe Casey?",
    ["(A) thankful", "(B) greedy", "(C) giving"])
```

Figure 3: Example CoGEX model-generated programs for Social IQa (Sap et al., 2019) questions. The left item fits well to a specific SocialIQa question pertaining to question-specific entities but does not generalize well to the dataset, while the right item applies more generally to cases such as the instance shown at the bottom, which does not pertain to character actions. Applying CoTACS to identify a single program such as the right one shows to improve overall task accuracy.

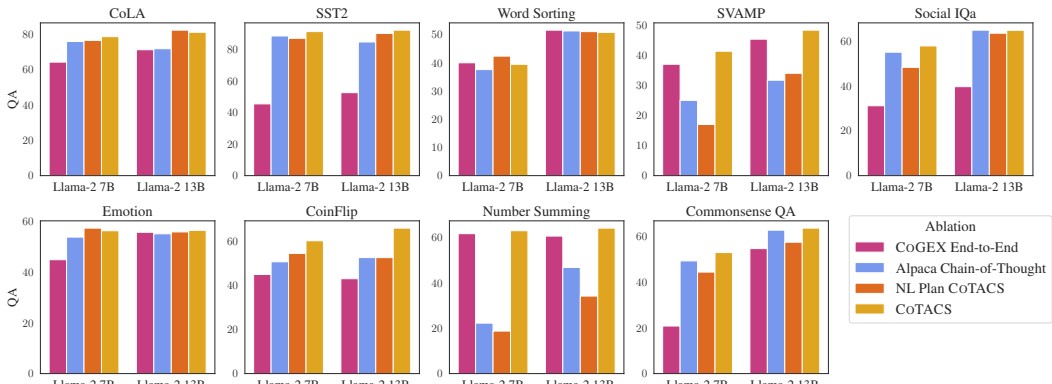

Figure 4: Performance comparison between CoTACS ($k$=1) and various ablations: (1) CoGEX End-to-End that generates separate programs for each instance, (2) chain-of-thought prompting, and (3) searching for an optimal NL plan instead of code program. CoTACS consistently equals or outperforms all ablations on all tasks, while each ablation drops in performance on at least 2-3 tasks.

**Is CoTACS better than chain-of-thought?** A common practice to elicit systematic reasoning from LMs is to prompt it for the reasoning via some version of "explain your answer step-by-step" (Kojima et al., 2022). How does this compare to CoGEX models on a given dataset? We compare CoGEX to zero-shot CoT by prompting our Alpaca models with a task-specific instruction, while additionally appending the instruction to "think step-by-step" before producing the answer. Figure 4 (blue vs gold) shows that CoT prompting performs similarly to the NL plan search method; it can approach CoTACS performance on some NLP classification tasks but performs worse over SVAMP, Number Summing, and CoinFlip.

**When is CoTACS better than fine-tuning?** Fine-tuning is a standard practice to adapt an LM to a new task. However, fine-tuning often requires a large amount of training data and storing a new model for each task. Here, we study the impacts of the number of examples on fine-tuning and CoTACS. We find that when there are many examples available, fine-tuning achieves stronger performance. However, CoTACS is generally better until there are a large number of examples available: it outperforms fine-tuning on 4/9 tasks with 500 examples. This suggests that CoTACS can be a lightweight alternative in the low-to-medium shot setup.

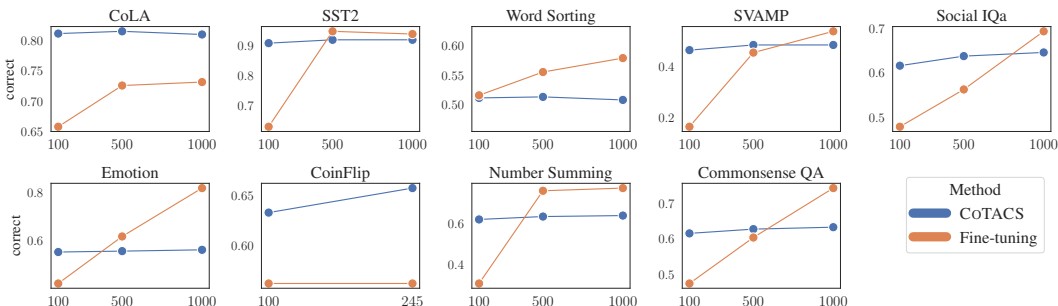

Figure 5: Performance tradeoff between CoTACS, which requires saving just a program string, and fine-tuning, which requires saving an entire checkpoint, as we increase the number of training examples. Although fine-tuning typically performs better with more data, CoTACS provides an alternative that is lighter-weight and stronger at low-to-medium numbers of instances.

```
def track_coin_flip(text, num_flip):
    """
    Track the current state of a coin after a certain number of flips.
    Args:
        text (str): a string containing information about the state of a coin at different stages.
        num_flip (int): the number of flips to consider.
    Returns:
        A dictionary containing (at least) the field 'answer', whose value is of type 'str' and contains
            the state of the coin after the specified number of flips.
        The dictionary also contains the result of the intermediate steps of the reasoning process.
    """
    # Step 1: Identify and extract all mentions of flipping a coin from the input text.
    coin_flip_mentions = extract_coin_flip_mentions(text)

    # Step 2: For each identified instance of flipping, determine whether it is a heads or tails outcome.
    flip_outcomes = {mention: determine_flip_outcome(mention) for mention in coin_flip_mentions}

    # Step 3: Track and record the state of the coin after each flip.
    coin_state = track_coin_state(flip_outcomes, num_flip)

    return {'coin_flip_mentions': coin_flip_mentions, 'flip_outcomes': flip_outcomes, 'answer': coin_state}

Example Input: 'A coin is heads up. Gee does not flip the coin. Joseluis flips the coin. Cory does not
    flip the coin. Stefanie flips the coin. Is the coin still heads up?'

Example Output: {
    'coin_flip_mentions': ['A coin is heads up', 'Joseluis flips the coin', 'Stefanie flips the coin'],
    'flip_outcomes': {
        'A coin is heads up': 'heads',
        'Joseluis flips the coin': 'heads',
        'Stefanie flips the coin': 'heads'
    },
    'answer': 'heads'
}
```

Figure 6: Qualitative examples of LLama-2 13B the coin flip tracking task where the model fails to correctly simulate the program and is correct for the wrong reasons.

### 3.4 Qualitative Analysis

Since we rely on the LM as a code emulator, there is no guarantee of correct execution. The generated intermediate outputs allow us to examine if the model can faithfully emulate the program. We observe failure cases where the LM incorrectly simulates the program execution even if the generated program is correct as shown in Figure 6. We also include positive qualitative examples in Appendix B.

## 4 Related Work

**Reasoning via Code.** Using code for reasoning is a burgeoning area that has shown improved results on many algorithmic tasks (Chen et al., 2022; Gao et al., 2023). Many approaches ask LLMs to express their reasoning as code and leverage code interpreters to execute them. Recently, and concurrent with our work, some studies investigate training LLMs as code compilers, where the LM is prompted to emulate code execution (Li et al., 2023; Chae et al., 2024; Mishra et al., 2023). These

LM-as-compiler approaches fall into a broader category of work that invokes LLMs as subroutines in programs (Kalyanpur et al., 2022; Weir et al., 2024). Different from ours, these works mainly rely on manually prompting very large models, while we focus on training open-source LMs to both generate and emulate programs. In addition, we aim to achieve task generalization by searching for an optimal program for a given task—different from Chae et al. (2024) who rely on prompting LMs with specific code instructions. Ours is the first work on code-based reasoning that employs search over the program space with the goal of generalizing an optimal program to a task.

**Prompt Optimization.** Our search procedure, CoTACS, has a similar spirit to in-context learning optimization approaches where the goal is to find an optimal set of exemplars (an optimal pseudo-program, in our case) for a given task. Existing studies (Zhang et al., 2022; Rubin et al., 2022; Ye et al., 2023a; Gupta et al., 2023b; Khalifa et al., 2023a) explored various methods to select optimal in-context examples, leveraging similarity- or diversity-based heuristics—to name a few. Searching for useful task instructions has also been explored (Honovich et al., 2022; Khalifa et al., 2023b; Chen et al., 2023).

Another related area of research is automated prompt engineering (Shin et al., 2020; Deng et al., 2022; Prasad et al., 2023) that bootstraps an effective prompt using some reward function. While LMs have been shown to be effective at producing their own prompts (Zhou et al., 2022; Yang et al., 2024; Pryzant et al., 2023; Ye et al., 2023b), our work shows that LMs can also reason by generating and executing their own generated programs. Our method differs from these studies as it uses the same input instruction and optimizes the intermediate representation, rather than modifying it via prompt optimization. Finding a single program string to solve a class of problems is also related to finding a high-level NL description of a task using one or multiple demonstrations (Weir et al., 2023).

# 5  Conclusion

We present CoGEX, a methodology that trains language models to generate and execute their own Pythonic reasoning programs in response to task instructions. We convert the Alpaca instruction tuning data into CoGEX instances that can be used to CoGEX-tune any models. We design an optimization algorithm, CoTACS, that applies CoGEX models to a new dataset by generating and searching through possible programs that can be reapplied to new task items. Applying the CoTACS search algorithm yields task performance that exceeds that of few-shot in-context-learning and typical NL instruction following. Our work demonstrates a way to apply LM-based programmatic reasoning to NLP benchmarks that require softer reasoning skills not easily stated in code syntax.

# 6  Limitations

While our work represents a step towards utilizing code language models for non-algorithmic reasoning tasks, CoGEX still suffers from the following limitations:

- CoGEX is suitable for soft reasoning tasks for which step-by-step programs are difficult to describe. However, when solving algorithmic tasks where a precise step-by-step program is possible, passing the generated code directly to an interpreter may be preferable to emulating code execution via the LM.

- We have found that the LM can occasionally produce a result that is inconsistent with the code emulated, as noted in subsection 3.4. In which case, the code does not faithfully reflect the reasoning process followed by the model.

- There is an extra computational overhead when emulating code execution via an LM compared to using an interpreter as the LM needs to generate intermediate variables along with the final answer.

# 7  Acknowledgements

We thank Li Zhang, Valentina Pyatkin, and Khyathi Chandu for feedback on ideas and earlier drafts. We also thank the organizers of the AI2 Summer 2023 Hackathon during which this project was initially conceived.

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

```
Clean up the following json items. Map the input and output fields in each item to a proper pythonic item
    (e.g. list, dictionary, or clean string). It shouldn't have newlines if it's a string. DO NOT INCLUDE
    "..." in your outputs.

INPUT 1:
{'instruction': 'Classify the following objects by color.', 'input': 'Ribbon, Tie, Pen', 'output': '-Red:
    Ribbon\n-Blue: Tie\n-Black: Pen'}

OUTPUT 1:
{'instruction': 'Classify the following objects by color.', 'input': ['Ribbon', 'Tie', 'Pen'], 'output': {'
    Ribbon': 'Red', 'Tie': 'Blue', 'Pen': 'Black'}

INPUT 2:
{'instruction': 'Convert the following text into a list.', 'input': 'The four elements of design are line,
    color, shape, and texture.', 'output': '- Line \n- Color \n- Shape\n- Texture'}

OUTPUT 2:
{'instruction': 'Convert the following text into a list.', 'input': 'The four elements of design are line,
    color, shape, and texture.', 'output': ['line', 'color', 'shape', 'texture']]}

INPUT 3:
{'instruction': 'Generate a list of five items a person might need for a camping trip', 'input': '', '
    output': '1. Tent\n2. Sleeping bags\n3. Flashlight\n4. Matches/lighter\n5. Insect repellent}

OUTPUT 3:
{'instruction': 'Generate a list of five items a person might need for a camping trip', 'input': '', '
    output': ['tent', 'sleeping bags', 'flashlight', 'matches/lighter', 'insect repellent']]}

INPUT 4:
{input}

OUTPUT 4:
```

Figure 7: Prompt used for converting inputs and outputs of Alpaca items into Pythonic data types.

```
Generate a high-level plan with at most 3 steps that a problem-solving artificial agent could use to
    complete the following problem. If the problem takes in inputs, your plan should be a high-level
    abstraction that is generally applicable to new inputs, not just the one shown here.

Instruction: {instruction}
Input: {possible_inputs}

Your output format should be a series of serialized jsons, 1 per line, for each step of the plan.
They should have the format {"number": <step number>", "description": <step description>}
```

Figure 8: Prompt used for generating stepwise NL plans for Alpaca items.

## A    Prompts for Converting Alpaca to CoGEX

Figure 7, Figure 8, and Figure 9 display the prompts used to GPT-4 in sequence to convert Alpaca into the CoGEX dataset. The prompts (1) convert all inputs and outputs into Pythonic types like strings, lists and dicts, (2) generate plans to answer a given instruction, and (3) instantiate each plan as a Python program with underspecified function calls.

## B    Further Qualitative Analysis

Figure 10 and Figure 11 show good qualitative examples generated by CoTACS, along with outputs from the 2-shot prompting baseline for the text classification and math reasoning tasks, respectively. We find that CoTACS encourages general-purpose code that is generalizable across multiple examples within the same task. It also enables better interpretability by generating outputs of the intermediate reasoning steps.

```
For the following questions with example inputs and outputs, generate a function that performs the
provided high-level steps.  The function should return a dictionary with the field "answer":  <answer>
as well as the values for intermediate decisions.  Don't hard code input-specific items whenever possible.
You can make external calls to undefined functions as long as the function name describes its purpose.

Instruction: Generate three antonyms for the word "wonderful".
Input:
Answer: ['horrible', 'abysmal', 'appalling']
Steps:
1. Search for synonyms of the target word using a thesaurus.
2. Identify antonyms of the synonyms found in step 3.
3. Package the antonyms as the output in the required format.
Code:
def generate_antonyms(num_words, word):
    """
    Generate antonyms for a given word.

    Args:
        num_words (int): the number of antonyms to generate for the word
        word (str): a string representing the word for which we need to find the antonyms.

    Returns:
        A dictionary containing the antonyms of the given word, plus the result of the intermediate steps
            of the reasoning process
    """

    # Step 1: Search for synonyms of the target word using a thesaurus.
    synonyms = thesaurus_lookup(word)

    # Step 2: Identify antonyms of the synonyms found in step 1.
    antonyms_of_synonyms = [lookup_antonyms(synonym) for synonym in synonyms]

    # Step 3: Package the antonyms as the output in the required format.
    all_antonyms = []
    for antonym_list in antonyms_of_synonyms:
        all_antonyms.extend(antonym_list)

    n_antonyms = all_antonyms[:num_words]

    return {
        'synonyms': synonyms,
        'antonyms_of_synonyms': antonyms_of_synonyms,
        'all_antonyms': all_antonyms,
        'answer': n_antonyms
    }

>>> generate_antonyms(3, 'wonderful')

Example Output:
output = {
    'synonyms': ['amazing', 'fantastic', 'terrific'],
    'antonyms_of_synonyms': [
        ['horrible', 'abysmal', 'appalling'],
        ['dull', 'disappointing', 'unexceptional'],
        ['awful', 'terrible', 'dreadful']
    ],
    'all_antonyms': [
        'horrible', 'abysmal', 'appalling', 'dull', 'disappointing',
        'unexceptional', 'awful', 'terrible', 'dreadful'
    ],
    'answer': ['horrible', 'abysmal', 'appalling']
}

###

Instruction: Generate ideas for a travel blog for young tourists visiting India

<...>

###

Instruction: {instruction}
Input: {input}
Answer: {output}
Steps:
{steps}
Code:
```

Figure 9: Prompt used for instantiating Python programs from NL plans. See repository for full-length prompt.

```
def determine_emotion(sentence):
    """
    Determine the emotion expressed in a given sentence.
    Args:
        sentence (str): the sentence for which the emotion is to be determined.
    Returns:
        A dictionary containing (at least) the field 'answer', whose value is of type 'str' and contains
            the emotion expressed in the sentence. The dictionary also contains the result of the
            intermediate steps of the reasoning process.
    """
    # Step 1: Extract all the words from the input sentence and analyze them to understand their context
        and meaning.
    words = extract_words(sentence)
    word_context = analyze_words_context(words)

    # Step 2: Identify the emotion-related words or phrases in the sentence.
    emotion_related_words = identify_emotion_related_words(words)

    # Step 3: Return the emotion that best fits the context and the emotion-related words identified in the
        sentence.
    emotion = determine_best_fit_emotion(word_context, emotion_related_words)

    return {'sentence': sentence, 'words': words, 'word_context': word_context,
            'emotion_related_words': emotion_related_words, 'answer': emotion}

Example Input: i was feeling festive yesterday

Example Output: {
    'sentence': 'i was feeling festive yesterday',
    'words': ['i', 'was', 'feeling', 'festive', 'yesterday'],
    'word_context': {
        'i': 'first person singular', 'was': 'past tense', 'feeling': 'verb', 'festive': 'adjective'
    },
    'emotion_related_words': ['festive'],
    'answer': 'joy'
}

2-shot Baseline Output: joy
```

Figure 10: Qualitative example of a LLama-2 13B CoGEX-generated program for the text classification task on the Emotion benchmark.

```
def solve_math_word_problem(word_problem):
    """
    Solve a math word problem.

    Args:
        word_problem (str): a string representing the math word problem.

    Returns:
        A dictionary containing (at least) the field 'answer', whose value is of type 'int' and contains
            the solution to the math word problem.
        The dictionary also contains the result of the intermediate steps of the reasoning process.
    """

    # Step 1: Identify the key numbers and variables from the problem statement.
    key_numbers = identify_key_numbers(word_problem)

    # Step 2: Understand the problem context.
    problem_context = understand_problem_context(word_problem)

    # Step 3: Perform the appropriate mathematical operations to solve the problem.
    solution = perform_math_operations(key_numbers, problem_context)

    return {
        'key_numbers': key_numbers,
        'problem_context': problem_context,
        'answer': solution
    }

Example Input: Paco had 36 cookies. He gave 14 cookies to his friend and ate 10 cookies. How many cookies
    did Paco have left?

Example Output: {
    'key_numbers': {'initial_quantity': 36, 'quantity_given_away': 14, 'quantity_eaten': 10},
    'problem_context': 'Paco had 36 cookies. He gave 14 cookies to his friend and ate 10 cookies.',
    'answer': 12}
}

2-shot Baseline Output: 20
```

Figure 11: Qualitative example of a LLama-2 13B CoGEX-generated program for the math reasoning task on the SVAMP benchmark.

