# OpenReview forum: "Learning to Reason via Program Generation, Emulation, and Search"
_NeurIPS.cc/2024/Conference — NeurIPS 2024 poster_

### Official Review · Reviewer_KtaJ · 2024-07-13

**Soundness:** 4
**Presentation:** 4
**Contribution:** 2
**Rating:** 6
**Confidence:** 4

**Summary:**

In this paper, the authors present a fine-tuned LLM that reasons by generating pythonic code and executes it through the model rather than passing it to an external interpreter (CoGEX). They also use the model to generate a set of candidate programs from a training set, among which the top-k performing programs are selected and subsequently used to evaluate the test set (CoTACS). The model shows improved performance on several datasets against benchmark models without the pretraining, including with in-context learning and chain-of-thought reasoning. Several additional analyses are provided for model efficacy, including the number of training items and sampled programs.

**Strengths:**

1. Paper is well-written and easy to follow. All the analyses and results are clearly presented.
2. The CoGEX model provides a useful datapoint on understanding how well LLMs can benefit from program-like planning without relying on an external program interpreter.
3. Allowing the model to use ‘primitive’ functions that are difficult to define programmatically offers greater flexibility than relying on interpreters and the ability to query for factual information.

**Weaknesses:**

1. The ideas do not strike me as particularly novel. As noted in the paper, using LLMs to generate code is not new, and neither is using language models to simulate program executors. CoTACS is simply a top-k search.
2. It isn’t clear to me what advantage this method has over just passing the code to Python Interpreter. Moreover, what advantage is there to using CoGEX to just using GPT4 given that the dataset was created using GPT4 in the first place?
3. While I appreciate the completeness of the presented results (for the sake of good science), even for experiments where the improvements are marginal, the results are often marginal at best, and it is unclear in what circumstances this approach is favorable and when it is not. Likewise, the authors use 0-shot CoT as their comparison model (Line 236), which is the weakest CoT baseline possible. Given that using an actual program interpreter is a viable alternative, I would be hesitant to adopt CoGEX without having a clearer understanding of when it outperforms simpler alternatives. In my opinion, the paper would strongly benefit from the following:
    A. A fair comparison with CoT, either controlling for the number of examples or showing at how many examples the models achieve parity. Since this is the simplest thing to try for most ML practitioners, even parity would have me favor CoT over most alternatives.
    B. A comparison, if applicable, to actual interpreters (e.g. Python). If this is not possible (e.g. due to calls to undefined functions), how often is that the case? Alternatively, if the model was forced to use predefined functions, how much better is it to use undefined functions?
    C. I really was hoping for a more extensive Qualitative Analysis section (S 3.4) to get a better understanding of when CoGEX strongly outperforms the baselines. The results in Table 1 demand too much prior knowledge about the individual tasks/datasets from the readers without some guidance on how to think about them.

**Questions:**

1. (Table 1): Could you define what BM25 means? It wasn’t clear to me from the paper exactly how you performed the benchmarks and how you chose the benchmark settings to ensure fair comparison.
2. (Line 91): What does silver-quality mean?
3. (Table 3 and Figure 2): It wasn’t clear from the paper whether the model generated meaningfully different programs. Footnote 3 (from Line 231) suggests that the model likely generated very similar outputs (assuming lower temperature means more deterministic outputs).
4. (Line 103): Few questions about ‘fuzzy’ premitives.
    A. Does the model come up with its own fuzzy primitive operations, or do they all need to be predefined by the engineer or GPT4?
    B. How did you determine what level of granularity was appropriate for the dataset?
    C. Does the model ever produce fuzzy primitives that were not included in the dataset by GPT4 and if so, how does the model deal with it?
5. (Line 187) Could you provide the actual SD or the 95% interval?
6. (Figure 2) Could you include baselines for these tasks, e.g. results from the off-the-shelf Llama-2 13B on these problems. (It’s hard to tell if the linear trends are misguiding for me or not because the x-axis is in log-scale while the y-axis is in raw percentages. Logit-scale for y-axis could potentially be more informative.)
7. (S3.4) Do you observe instances where the generated program is incorrect but the model still gets the problems correct? In other words, does the model actually follow the incorrect program or does the model ignore it?

**Limitations:**

The authors provide no guarantee of safety (this is acknowledged in the supplements), but not addressed.

---

> ### Author Rebuttal · Authors · 2024-08-07
>
> We greatly appreciate your feedback on our draft. We are excited that you have found our approach useful and flexible. We aim to address your concerns below:
>
> > W1: The ideas do not strike me as particularly novel. As noted in the paper, using LLMs to generate code is not new, and neither is using language models to simulate program executors. CoTACS is simply a top-k search.
>
> This work shows how a code execution approach can be applied to a significantly broader class of tasks than previously considered. Generating code with LMs is indeed not novel, but to date has been limited to algorithmic tasks that can be easily expressed programmatically. **Our key contribution is to extend this approach to more complex tasks where a full, programmatic implementation is difficult or infeasible** (e.g., requires commonsense). Our novel solution is to generate partial (pseudo-)programs: while these cannot be directly executed, we find that LLMs can simulate their execution, including filling in gaps for calls to complex functions that would have been difficult to implement in code.
>
> > W2: It isn’t clear to me what advantage this method has over just passing the code to Python Interpreter.
>
> Since the generated code represents pseudo-programs that include undefined functions, **passing the code to an interpreter will not work**. Therefore, we rely on the LLM to emulate code execution. As for using GPT-4, it can indeed be used to synthesize pseudo-programs at test time, but also can smaller, less expensive LLMs as we show. Whether GPT-4 can produce more expressive pseudo-programs at test time compared to our fine-tuned models is unclear and would need additional experiments to verify. The relatively cheap inference of smaller models enables us to perform our program search to find an optimal pseudo-program for a given dataset/task.
>
> > W3.1: the results are often marginal at best.
>
> CoGEX shows substantial improvements compared to the baselines over most of the tasks we consider e.g., CoLA (+8.6), CoinFlip (+12.4), WSorting (+6.1), CSQA (+18.2), etc. Our results highlight how CoGEX can provide an effective alternative to the baselines for reasoning such as vanilla CoT.
>
> > W3.2  the paper would strongly benefit from a fair comparison with CoT..
>
> The main obstacle here is that all the datasets we experiment with do not provide ground-truth CoT in their training data. However, our 2-shot BM25 baseline is a fair comparison since it has access to the same number of examples as CoGEX.
>
> > W3.3 . B. A comparison, if applicable, to actual interpreters.
>
> It would not be possible to use actual interpreters, since almost all the generated pseudo-programs include at least one call to an undefined function. In addition, many of the tasks we evaluate on such as Social QA and Commonsense QA involve soft reasoning that can not be described in code.
>
> > W3.4 if the model was forced to use predefined functions, how much better is it to use undefined functions?
>
> Predefined functions require precise reasoning, while our work tackles soft reasoning that cannot be easily described in code.
>
> > W3.5 C. A more extensive Qualitative Analysis section (S 3.4) to get a better understanding of when CoGEX strongly outperforms the baselines.
>
> We include a high-level overview of the tasks/datasets we cared about in L147-160. We will add extra info and examples from each dataset in the Appendix to help with this in the revision. Also, based on your suggestion, we will include a more detailed qualitative analysis of CoGEX vs. the baselines.
>
> > Q1: (Table 1): Could you define what BM25 means?
>
> BM25 is a ranking algorithm which we use to retrieve similar examples from the training data to use as in-context examples. Retrieval is common to boost in-context learning by using examples that are similar to the input as demonstrations [[1](https://arxiv.org/abs/2101.06804)]
>
> > Q2: Line 91): silver-quality
>
> "Silver quality" typically refers to data that is of good but not perfect quality. It is a step below "gold quality" data, which is considered to be the highest quality and typically manually annotated by human experts.
>
> > Q3: Footnote 3 (from Line 231) suggests that the model likely generated very similar outputs (assuming lower temperature means more deterministic outputs).
>
> ​​In L231, we say that we sample with temperature=0.05 only with the end-to-end model. With search, we use T=0.7 as indicated in L165.
>
> > Q4.1: Does the model come up with its own fuzzy primitive operations?
>
> Indeed, after training, the model learns to generate its own primitives based on the input.
>
> > Q4.2: How did you determine what level of granularity was appropriate for the dataset?
>
> When constructing our training data, GPT-4 handles determining the appropriate granularity for the Alpaca data. Our trained models then learn to generate programs of appropriate granularity based on the input instance. We further apply search to find an optimal program (and granularity) for a given task.
>
> > Q4.3: C. Does the model ever produce fuzzy primitives that were not included in the dataset by GPT4?
>
> Certainly, the model learns to generate appropriate primitives based on the provided task instance. Based on our observations and analysis, the LM can generate and emulate novel primitives not seen in the training data.
>
> > Q5: (Line 187) Could you provide the actual SD or the 95% interval?
>
> We will add this information in the revision.
>
> > Q6: (Figure 2) Could you include baselines for these tasks, e.g. results from the off-the-shelf Llama-2 13B?
>
> We will add a line to the plots to show the baseline performance for each task.
>
> > Q7: (S3.4) Do you observe instances where the generated program is incorrect but the model still gets the problems correct?
>
> Yes, this sometimes happens, especially for tasks with binary output such as Coin Flip. However, we observe that the model mostly follows the program logic, which we verified by looking at the generated intermediate outputs.

---

### Official Review · Reviewer_Hd75 · 2024-07-14

**Soundness:** 2
**Presentation:** 3
**Contribution:** 2
**Rating:** 5
**Confidence:** 3

**Summary:**

The proposed approach, Code Generation and Emulated Execution (COGEX), involves training LMs to generate pseudo-programs with some undefined functions and then emulate the execution of these programs. This allows the LM's knowledge to fill in the gaps while maintaining a program like reasoning framework. The COGEX model is adapted to new tasks through program search, optimizing performance across dataset instances. The approach shows improvements over standard in-context learning methods, demonstrating that code synthesis can be applied to a wider range of problems.

**Strengths:**

- The concept of pseudo-program and LLM emulation of program is interesting.
- The authors evaluated the methodology on many different dataset and tasks.

**Weaknesses:**

I would consider raising my score if a fair number of the following weaknesses can be fixed by the authors. Many of them should be doable without extra experiments.

- Motivation: While in general I like the idea of pseudo-programs, but I don’t think it is very well motivated or explained in the paper. For this relatively new concept to be introduced, I would like to see a more precise definition that describes each type of reasoning, whether its intuition based with no reasoning, “soft reasoning”, and exact reasoning. State clearly what task falls into the camp of “soft reasoning”.
- Examples: I think the motivation is insufficient partly because of that the motivating examples (Fig. 1, 3, 10) do not seem to be requiring any “complex reasoning”. By “complex reasoning” I specifically mean things like composition of multiple sources of information, long range dependency of information, the use of logical operations like “and”, “or”, “not”, and “implies”, and etc.
- Task/Dataset selection: (also related to previous points) I would want some more clarification on why the authors pick the presented tasks and datasets. What makes these dataset special so that COGEX can be applied to them? What additional conceptual benefits do COGEX bring to them? This will also help motivate the methodology you proposed.
- Formalization: the symbol $f$ for COGEX model is unnecessarily abused. It can be invoked with either 2 or 3 arguments, which does not type check as a well-defined formalization. You should at least create two separate symbols, such as $f_{\text{reasoner}}$ and $f_{\text{enumlator}}$.
- Algorithm: The while loop in line 7-10 of Algorithm 1 does not seem to have termination guarantee. This is a potential flaw in the algorithm. What if the task and dataset is malformed and there exists no proper pseudo-program? Your algorithm should account for this.
- Comparison to GPT-4: I don’t want to suggest to do new experiments with GPT-4 as baselines, but I do want clarification on why GPT-4 is not used as a zero-shot baseline. I see that authors use GPT-4 to generate datasets used for training. So it’s definitely not the case that authors do not have access or hardware/financial limitations for doing so. Then why isn’t GPT-4 used as a baseline? I would assume that GPT-4 without training can still be used to synthesizing pseudo-programs, right?
- Claim in the experimental results: authors describes Fig.4 as “Figure 4 (blue vs gold) shows that CoT… can approach COTACS performance on some NLP classification tasks but generally performs substantially worse.” But to me when looking at Fig.4, I don’t see this claim accurate. I can only see significant difference on CoinFlip and Number Summing.
- Experimental setting: (related to the previous point) the direct comparison with zero-shot CoT might not be fair because your model has undergone fine-tuning, right?

**Questions:**

(I asked majority of questions along with the listed weaknesses)
- Typo in Figure 3 caption: “SocialIQa” -> “Social IQa”

**Limitations:**

- The author listed the limitations in the Appendix, which I would prefer doing in the main text. This also relates to the first listed weakness that I wrote: giving a proper overview would help people identify where and when your methodology could be applied.

---

> ### Author Rebuttal · Authors · 2024-08-07
>
> Thank you for your elaborate review and feedback on our submission. We aim to address your concerns below:
>
> > Motivation: While in general I like the idea of pseudo-programs, but I don’t think it is very well motivated or explained in the paper. For this relatively new concept to be introduced, I would like to see a more precise definition that describes each type of reasoning, whether its intuition based with no reasoning, “soft reasoning”, and exact reasoning. State clearly what task falls into the camp of “soft reasoning”.
>
> We define pseudo-programs in L7-8 and L29-34, mainly as programs with one or more leaf functions left undefined. While ours is not a formal definition, we believe it should be sufficient for the scope of our paper. As for “soft reasoning”, it refers to tasks whose solution is difficult to fully describe in code. We agree with your feedback here, and we intend to give a more concrete definition of it in future revisions.
>
> > Examples: I think the motivation is insufficient partly because of that the motivating examples (Fig. 1, 3, 10) do not seem to be requiring any “complex reasoning”. By “complex reasoning” I specifically mean things like composition of multiple sources of information, long range dependency of information, the use of logical operations like “and”, “or”, “not”, and “implies”, and etc.
>
> Many of the tasks we work with require a level of multi-hop reasoning. For example, solving SVAMP requires multistep aggregation of math operations, as in the following example:
>
> ```
> Robin has 28 packages of gum and 14 packages of candy. There are 6 pieces in each package. How many pieces does Robin have?
>
> Solution: ( ( 28.0 + 14.0 ) / 6.0 )
> ```
>
> Also, CSQA requires understanding multiple commonsense relations between entities as in the following entities:
> ```
> The teacher told all the students that listening was key, it was the main way they would gain what? [ "empathy", "anxiety", "knowledge", "falling down", "hear things"]
>  ```
> You are right that we didn’t consider tasks targeting e.g., discrete logical operations. However, the **key contribution of our work is not to push the boundary of complex reasoning that LLMs can do, but to show that code generation can be utilized to solve NLP tasks whose solution cannot easily be described in code**, and we show on a variety of standard reasoning benchmarks that are commonly used in the literature.
>
>
> > Why the authors pick the presented tasks and datasets. What makes these dataset special so that COGEX can be applied to them? What additional conceptual benefits do COGEX bring to them?
>
> We chose these particular datasets to get a representative sample of different categories of NLP tasks currently of interest to the community (e.g. typical classification like SST/emotion, commonsense QA tasks (CSQA, SIQA), math (SVAMP) etc. There is nothing particularly “special” about these datasets except for the fact that a shared "plan," expressible as a discrete pseudo program, can be readily applied to all the different data points for the given task.  We drew a number of them from recent work on code generation, e.g. the text classification tasks from [[1](https://aclanthology.org/2024.findings-naacl.259.pdf)].
>
> > Formalization: the symbol 𝑓 for COGEX model is unnecessarily abused. [...] You should at least create two separate symbols, such as 𝑓reasoner and 𝑓emumlator
>
> Thanks for pointing this out; we agree that it would be clearer to create two separate symbols and will modify the paper accordingly
>
> > Algorithm: The while loop in line 7-10 of Algorithm 1 does not seem to have termination guarantee. This is a potential flaw in the algorithm. What if the task and dataset is malformed and there exists no proper pseudo-program? Your algorithm should account for this.
>
> Thank you for this insightful observation. We apologize for overlooking this case and will modify the algorithm accordingly in the revision.
>
>
> > why isn’t GPT-4 used as a baseline? I would assume that GPT-4 without training can still be used to synthesizing pseudo-programs, right?.
>
> Our concept of pseudo-programs is agnostic to the synthesizer model used. Indeed, GPT-4 can be used to synthesize pseudo-programs at test time, but also can smaller, less expensive LLMs as we show. Whether GPT-4 can produce more expressive pseudo-programs at test time compared to our fine-tuned models is unclear and would need additional experiments to verify. In addition, the relatively cheap inference of smaller models enables us to perform our program search during training to find an optimal pseudo-program for a given dataset/task.
>
> > When looking at Fig.4, I don’t see this claim accurate. I can only see significant difference on CoinFlip and Number Summing.
>
> Thank you for the observation. We apologize for this mistake, and we will revise to make it clear that CoTACS substantially outperforms CoT on Summing, SVAMP, and CoinFlip.
>
> > Experimental setting: (related to the previous point) the direct comparison with zero-shot CoT might not be fair because your model has undergone fine-tuning, right?
>
> Zero-shot CoT refers to the way the model is prompted i.e., without in-context examples and using “Let’s think step-by-step” in the instruction. But just like our models are fine-tuned on the CoGEX data, the baseline is finetuned on the original Alpaca dataset before prompting. So the comparison is fair.
>
> > The author listed the limitations in the Appendix, which I would prefer doing in the main text. This also relates to the first listed weakness that I wrote: giving a proper overview would help people identify where and when your methodology could be applied.
>
> Thank you for this observation. This was due to the page limit, but we will make sure to move the limitations section to the main text as you suggested.
>
>
> We hope we have addressed all your concerns. Please let us know if you have any more concerns or questions.

---

> > ### Comment · Reviewer_Hd75 · 2024-08-13
> > **Thanks for the Rebuttal**
> >
> > Thank you for your rebuttal. While the response cleared up some of my concerns, there are still many confusion left, such as the motivation of pseudo programs and the scientific definition of exact and soft reasoning. I’m keeping my current rating as is.

---

> > > ### Author Response · Authors · 2024-08-13
> > > **Thanks for the response**
> > >
> > > Thank you for your response!
> > >
> > > Could you give a few more specifics about your remaining points of confusion following our responses in the rebuttal? Hopefully we can clarify them during the remaining time of discussion period.

---

> > > ### Author Response · Authors · 2024-08-13
> > > **Clarification about pseudo-programs and soft reasoning**
> > >
> > > If it is helpful, here is a more specific comparison between the tasks we consider with pseudo-programs vs what previous works have done with regular programs.
> > >
> > > The PAL (Program-aided Language Models) paper ([Gao et al 2023](https://arxiv.org/pdf/2211.10435)) uses Codex to generate fully executable Python programs to solve tasks in the following categories: Math reasoning, Symbolic reasoning, Algorithmic reasoning.
> > >
> > > These involve tasks like GSM, where the program looks like this:
> > > ```
> > > Q: Olivia has $23. She bought five bagels for $3 each. How much money does she have left?
> > >
> > >
> > > money_initial = 23
> > > bagels = 5
> > > bagel_cost = 3
> > > money_spent = bagels * bagel_cost
> > > money_left = money_initial - money_spent
> > > answer = money_left
> > > ```
> > >
> > > The task thus boils down to 1. Parsing the problem into symbolic variable assignments, then 2. applying python-implemented internal functions like addition and subtraction.
> > > This is the case for all the kinds of tasks they consider for PAL. E.g. for their algorithmic tasks like OBJECT COUNTING:
> > > ```
> > > # Q: I have a chair, two potatoes, a cauliflower, a lettuce head, two tables, a cabbage, two onions, and three fridges. How many vegetables do I have?
> > >
> > >
> > > vegetables_to_count = {
> > > 'potato': 2,
> > > 'cauliflower': 1,
> > > 'lettuce head': 1,
> > > 'cabbage': 1,
> > > 'onion': 2
> > > }
> > > answer = sum(vegetables_to_count.values())
> > >
> > > ```
> > > The LM thus serves as a parser of the problem into a setting in which the solution falls out based on symbolic manipulation operations that Python easily handles. Thus, our definition of “exact reasoning” tasks are those for which a simple solution recipe contains only definitions and well-formed Pythonic operations in order to aggregate the definitions into an answer.
> > >
> > > Now let’s consider some of the tasks we look at in our paper, like the CSQA problem from our rebuttal:
> > > ```
> > > The teacher told all the students that listening was key, it was the main way they would gain what?  (A) "empathy" (B) "anxiety" (C) "knowledge" (D) "falling down" (E) "hear things"]
> > > ```
> > > What is the corresponding PAL program that could solve this question? It requires considering each option and the possible common sense relations between them and the situation in the question. One could code up a query to a symbolic ontology, but this is a much more complex solution than just generating code from an LM and feeding it to a Python executor. Instead, our tactic is to let the LM generate a pseudo-program that doesn’t actually get executed; it suggests to the LM a programmatic means to solve the CSQA instance, but circumvents the hard-to-code parts of the solution via underspecified function calls.  Since it is underspecified, we use the LM to emulate the program execution because Python would not be able to.
> > > ```
> > > # Step 1: Extract the main statement and options from the question.
> > > statement, options = extract_statement_and_options(question)
> > > # Step 2: Identify the key action and its purpose in the statement.
> > > key_action, purpose = identify_key_action_and_purpose(statement)
> > > # Step 3: Match the purpose with the most relevant option.
> > > answer = match_purpose_with_option(purpose, options)
> > > return {'statement': statement, 'key_action': key_action, 'purpose': purpose, 'answer': answer}
> > > ```
> > >
> > > Each of these functions like ‘identify_key_action_and_purpose’ are undefined– they would be very hard to actually code up in pure Python.  We face the same sort of challenge for tasks like sentiment analysis.
> > >
> > > Thus, by “soft reasoning” we refer to NLP tasks that require nontrivial reasoning steps that you can’t write simple Python snippets to handle, such as ‘identify_key_action_and_purpose’.
> > > We described this idea in L29-34 of the submission, but will expand it to make this more clear to future readers. As we state in the paper and in our rebuttal, ours is the first paper to handle these sorts of soft reasoning tasks using a code-based approach.
> > >
> > > We hope this helps clear up your confusion. Please let us know if not, we are happy to do so while discussion period is still open.

---

### Official Review · Reviewer_a6dH · 2024-07-15

**Soundness:** 3
**Presentation:** 3
**Contribution:** 3
**Rating:** 7
**Confidence:** 4

**Summary:**

This paper proposes a means of training language models to perform semi-programmatic reasoning, along with an adaptation method based on program search to specialize the resulting models for particular downstream tasks without updating their parameters.

The authors use GPT-4 to generate programmatic reasoning strategies from Alpaca instruction-tuning examples, allowing GPT-4 to include undefined function calls as the resulting code does not actually need to be executable, only used to guide the downstream language model's inference process.

The authors evaluate their method by fine-tuning Llama 2 models on their version of the Alpaca dataset, then comparing benchmark performance against few-shot, instruction-tuned zero-shot, and chain-of-thought baselines on several tasks picked to represent both algorithmic and non-algorithmic reasoning.

Experimental results indicate that the proposed method outperforms the considered baselines across most tasks by a good margin.

**Strengths:**

- The proposed method appears to strike a nice balance between the rigidity of code-structured reasoning and freeform CoT, as it performs well across domains where CoT excels and ones where it struggles.
- Being able to specialize reasoning for a particular task by selecting a pseudo-program is neat. It also appears to work well with a much smaller number of examples than are required to effectively fine-tune a model.
- The authors do a very good job of proactively answering natural questions with their ablations in section 3.3.
- As for clarity of exposition, all the writing is easy to follow and the paper is laid out intuitively.

**Weaknesses:**

1. The authors don't report statistical significance (e.g. through bootstrapping) or variance across runs with different subsamplings of training data.
2. As far as I can tell, the reported experiments don't really include domains where the correct solution can be reached by actually executing a fully specified program, besides Number Summing (accordingly, a program-of-thought baseline is also missing). In CoGEX the model is responsible for simulating program execution, so it seems likely that it would underperform in these settings. While actually integrating an interpreter into a system that can still handle underspecified functions would be out of scope, it would be good to address this issue at the conceptual level somewhere in the text.

**Questions:**

1. Did you try few-shot with the Alpaca (instruction-tuned) models that you used zero-shot in the paper?

**Limitations:**

The authors include a short statement to the effect that their models may make reasoning mistakes and the model artifacts are provided without any guarantees of safety. It might be worth including the fact that models may fail to execute generated code consistently, and thus generated code may be seen as an unfaithful explanation of model reasoning.

---

> ### Author Rebuttal · Authors · 2024-08-07
>
> We sincerely thank you for the constructive feedback on our submission. We are glad that you found our approach neat, our ablations useful, and our writing clear. We aim to address your concerns below.
>
> > W1: The authors don't report statistical significance (e.g. through bootstrapping) or variance across runs with different subsamplings of training data.
>
> Thank you for pointing this out, due to the limited time of the rebuttal, we will compute statistical significance of CoGEX against the baselines and add variance information to Figure 2 in the revision.
>
> > W2: As far as I can tell, the reported experiments don't really include domains where the correct solution can be reached by actually executing a fully specified program, besides Number Summing (accordingly, a program-of-thought baseline is also missing).
>
> Thank you for this insight. As there is already much literature about generating correct programs (e.g., program-of-thought, PAL, etc.), we want to focus on tasks where this is less feasible. We therefore focus on tasks where it is hard to describe the reasoning in code (see line number L29-L32) and which will not benefit much from an interpreter. We will highlight the pros and cons of CoGEX compared to using an actual interpreter in the revision, as you suggested.
>
> > Q1: Did you try few-shot with the Alpaca (instruction-tuned) models that you used zero-shot in the paper?
>
> We did, but found it to perform significantly worse than zero-shot. This is mainly due to that instruction tuning is zero-shot, i.e., the model is trained to generate output given the instruction and input without in-context examples. We will include a note describing this in the next version of the paper.
>
> > It might be worth including the fact that models may fail to execute generated code consistently, and thus generated code may be seen as an unfaithful explanation of model reasoning.
>
> Thank you for noting this. We will include this in our limitations in the paper revision.

---

### Author Response · Authors · 2024-08-12
**Nearing the end of discussion period**

We thank all reviewers for taking the time to carefully consider our paper and give valuable constructive feedback. We have provided responses to reviewer criticisms below on a case-by-case basis.

We have not yet heard from reviewers during the author-reviewer discussion phase. Before it ends tomorrow (August 13th 11:59PM AoE), we would appreciate it if any reviewers would like to comment as to whether we have addressed their concerns, or if there are still remaining issues that require further discussion. Otherwise, we’d ask that reviewers consider changing their score in light of our clarifications.

Thanks!

---

### Decision · Program_Chairs · 2024-09-25

**Decision:**

Accept (poster)

**Comment:**

Thanks to the authors for submitting this work and engaging in the rebuttal period. I found that the paper introduced an innovative approach that balances code-based reasoning and thought-style reasoning by adopting more pseudo-program structures. This is a more relaxed approach to facilitate better reasoning with LLM while not requiring fully specified programs and corresponding program interpreters. The authors conducted comprehensive experiments and reported strong results with insightful ablation analysis.

As noted by the reviewers, there are some minor issues found: (1) the approach might not perform as well in domains where exact/hard reasoning steps are more effective i.e. where the actual execution of a full program will lead to the correct output; (2) second, some experimental results demonstrate only marginal performance gains of the proposed method.

Overall, I recommend accepting this paper and hope the authors will continue to improve the paper according to the above-mentioned points and other feedback from reviewers.